# Hardware Realization of the Pattern Recognition with an Artificial Neuromorphic Device Exhibiting a Short-Term Memory

**DOI:** 10.3390/molecules24152738

**Published:** 2019-07-28

**Authors:** Dawid Przyczyna, Maria Lis, Kacper Pilarczyk, Konrad Szaciłowski

**Affiliations:** Academic Centre for Materials and Nanotechnology, AGH University of Science and Technology, al. Mickiewicza 30, 30-059 Kraków, Poland

**Keywords:** photoelectrochemistry, wide bandgap semiconductor, artificial neuron, in materio computing, neuromorphic computing

## Abstract

Materials exhibiting memory or those capable of implementing certain learning schemes are the basic building blocks used in hardware realizations of the neuromorphic computing. One of the common goals within this paradigm assumes the integration of hardware and software solutions, leading to a substantial efficiency enhancement in complex classification tasks. At the same time, the use of unconventional approaches towards signal processing based on information carriers other than electrical carriers seems to be an interesting trend in the design of modern electronics. In this context, the implementation of light-sensitive elements appears particularly attractive. In this work, we combine the abovementioned ideas by using a simple optoelectronic device exhibiting a short-term memory for a rudimentary classification performed on a handwritten digits set extracted from the Modified National Institute of Standards and Technology Database (MNIST)(being one of the standards used for benchmarking of such systems). The input data was encoded into light pulses corresponding to black (ON-state) and white (OFF-state) pixels constituting a digit and used in this form to irradiate a polycrystalline cadmium sulfide electrode. An appropriate selection of time intervals between pulses allows utilization of a complex kinetics of charge trapping/detrapping events, yielding a short-term synaptic-like plasticity which in turn leads to the improvement of data separability. To the best of our knowledge, this contribution presents the simplest hardware realization of a classification system capable of performing neural network tasks without any sophisticated data processing.

## 1. Introduction

Pattern recognition is one of the basic cognitive functions, which, due to its complexity and required accuracy, has challenged researchers for decades in a strive to mimic it in an artificial setup. The development of such systems is fueled by various possible applications in medicine, security, economics and many other fields of human’s activity. At the moment, the majority of available solutions are based on various software implementations of the machine learning approach, including above all the use of artificial neural networks (ANN) of different architectures. In most of the cases, the ANNs principle of operation is based on the optimization of weights associated with individual connections between nodes (neurons) and the information flow is inspired by the functions of biological structures found in the nervous system. It has been proven on numerous occasions that these algorithms provide an excellent efficiency in various classification tasks with both supervised and unsupervised learning procedures [1,2,3].

In spite of this, the use of software implementations for ANN algorithms often requires heavy pre- and post-processing of the analysed data and/or a high degree of network complexity translating into a high energy consumption [4]. The use of ANN-based methods may also be associated with a potentially low tolerance towards deliberate attacks [5], emphasized in the case of the one-pixel attack capable of deceiving certain deep neural networks [6]. To meet the discussed drawbacks, some researchers propose the development of hardware implementations for neural networks architectures, incorporating novel materials, non-classic electronic elements and unconventional computing paradigms, such as multi-valued and fuzzy logic systems [7].

The development of neuromorphic computing (NC) can be perceived as one of the manifestations of this trend. The key idea here is to design a brain-inspired hardware computing platform which is optimized towards the implementation of selected aspects of ANN algorithms [7,8]. Among the most advantageous concepts, the use of circuitry employing spiking artificial synapses has been proven to be more energy efficient than the implementation in silico [8]. The construction of these systems is currently being investigated in terms of new materials [9,10,11] which are applied within integrated networks capable of performing sophisticated information processing [12,13,14]. At the same time, a number of studies aim at simplifying the circuitry realizing the neuromorphic computations. Wang et al. [15] demonstrated a hybrid convolutional neural network with only one spiking synapse based on a HfO_2_ memristor. The system was capable of recognizing handwritten digits with the efficiency of 784 neurons. It was achieved through the time division multiplexing access technique. Nonetheless, the “network” required multilayer information pre-processing and several thousands of software neurons to operate.

The research on photochemical and photoelectrochemical in materio computing devices indicates the possibility of their integration into larger computing systems with the use of optical [16,17] or electrical [18] signals. This can lead to the construction of more complex photoelectrochemical circuits [18], molecular arithmetic-logic units [17] or molecular-scale neural networks [16] and communication systems [19] capable of the sophisticated information processing. The studies on artificial photoelectrochemical synapses [20], devices that may realize elementary learning processes (e.g., paired pulse facilitation, PPF), stimulated further development of the neuromorphic systems combining the neuromimetic approach towards data processing with in materio computing concepts [21,22,23,24,25]. The operation principle of a photoelectrochemical synapse is based on the competition between light-induced charge carrier generation, charge carrier trapping and other interfacial processes affecting the photocurrent generation. Whereas the information processing realized with the use of unconventional molecular or nanoscale devices has several drawbacks compared to classic, silicon-based electronics (usually low speed, some problems with data encoding and concatenation) [26], its combination with classic techniques and algorithms seems to be promising. The term heterotic computation encompasses hybrid systems, in which information processing is performed on various platforms depending on the optimal scenario, utilizing the speed and maturity of in silico computing or the flexibility of the unconventional approaches [27,28].

Here, we present an extremely simplified, robust circuit made of only one photoelectrochemical element, the operation of which is similar to a simple classification system. In the discussed case an Modified National Institute of Standards and Technology Database (MNIST) set of handwritten digits serves as the input data under consideration [29]—without the use of any data pre-processing or software ANNs. The presented optoelectronic device realizes the paired-pulse facilitation (PPF)—a type of short-term plasticity (STP)—seen as an enhancement of the postsynaptic current resulting from the increase in stimulating events frequencies [30,31]. Therefore, we are testing the information processing capability of a single artificial neuron made of nanocrystalline cadmium sulfide. The obtained results show, that the use of a such simple system may improve the separation in the phase space based solely on the characteristics of the input data (unsupervised learning). It seems possible that the proposed approach could be scaled up and a network of similar, interconnected devices could serve as a complex hardware neural network implementing the fuzzy logic formalism and selected concepts of reservoir computing [32].

## 2. Results and Discussion

### 2.1. Material Characterization

In order to determine the band gap width (E_g_) of cadmium sulfide (CdS), the reflectance spectrum was recorded. Kubelka–Munk’s function F_KM_ was calculated based on the raw data and a Tauc plot was made (Figure 1a). CdS is typically considered a direct semiconductor with the E_g_ value equal to 2.42 eV for the hexagonal phase and 2.33 eV for the tetragonal polymorph. The value determined for the discussed material (2.33 eV, Figure 1a) may suggest the dominance of the latter polymorph, but this value is usually also observed for mixtures of both crystalline phases [33].

The powder X-ray diffraction measurements have been employed to assess the CdS sample composition. The obtained data was analyzed using HighScore Plus software [34] in which so called Rietveld refinement was applied [35]. This method allows evaluation of certain parameters including the volume fraction of phases. The analysis conducted for the diffraction pattern shown in Figure 1b indicates that both the tetragonal hawleyite [36] and hexagonal greenockite [37] phases are present in approx. 1:1 volume ratio. The energy dispersive X-ray spectroscopy (Figure 1c) indicates the absence of significant impurities, therefore electronic trap states (vide infra) most likely originate from CdS lattice defects. The SEM image (Figure 1d) reveals heavily agglomerated material, for which the particle size statistics were calculated using Image J software [38]. The distribution of crystalline diameters is relatively narrow, ranging from 25 to 110 nm, with an average diameter of 71 ± 3 nm, whereas the distribution maximum is found at 53 ± 1 nm.

### 2.2. Plasticity of the Artificial Neuron

The composites of cadmium sulfide with multiwalled carbon nanotubes reported in our previous works exhibited memory features that can be functionally associated with the neuronal facilitation (particularly, the paired-pulse facilitation—PPF) in terms of the short-term synaptic plasticity [20]. In the neuroscience, the PPF is considered a neuronal enhancement mechanism which consists of four distinctive processes characterized by different time constants and different physiology [39]. The PPF causes an amplification of the postsynaptic response as a consequence of the increase in stimulating event frequencies at the presynaptic axon [40]. It is believed that the PPF is realized mainly through the accumulation of depolarizing Ca^2+^ ions in the presynaptic neuron [30]. High frequency components of the PPF mechanism (i.e., these characterized by low time constant values) may be useful from the point of view of information processing. These include the fast-decaying facilitation F1 and the slow-decaying facilitation F2 [30,39,41]. The influence of both components manifests itself in the double exponential decay of the postsynaptic response depicted in Figure 2.

A similar phenomenon was observed for the photoelectrodes made of nanocrystalline cadmium sulfide (Figure 3). Like in the case of multiwalled carbon nanotubes (MWCNT)-CdScomposite [20], when the interval between light pulses is sufficiently long (over 300 ms for this study) the subsequent photocurrent spikes are unafected by previous states the device was in. However, if the interval between irradiations becomes shorter (e.g., 80 ms) the amplifiation of the second photocurrent response becomes significant. Detailed analysis reveals that the ratio of pulse intensities (the facilitation rate) vs. the time interval between stimuli is best fitted with a biexponential function (1):(1)$$\frac{A_{2}}{A_{1}}=a_{1}e^{-\frac{t}{\tau_{1}}}+a_{2}e^{-\frac{t}{\tau_{2}}}+y_{0}$$
which is fully consistent with the previous reports on MWCNT-CdS composite photoelectrodes [20]. The result of the fitting procedure is presented in Figure 3b and the parameters equal to: *a*_1_ = 0.218 ± 0.023, *a*_2_ = 0.340 ± 0.014, *τ*_1_ = 19.8 ± 4.6 ms, *τ*_2_ = 167 ± 23 ms and *y*_0_ = 1.008 ± 0.013. The time constants values, which are representative for polycrystalline CdS samples, are slightly higher than those obtained for CdS/MWCNT composites [20]. Interestingly, the determined values are consistent with the parameters typically observed in the case of biological structures [31].

The double exponential decay can be associated with two distinctive trapping/detrapping events characterized by two time constants τ_1_ and τ_2_. This diversity may originate from the presence of two CdS polymorphs, of which charge trapping states most likely differ. At the same, through the comparison with selected natural learning processes, these two mechanisms may be associated with two components of neuronal plasticity: the fast-decaying facilitation F1 and the slow-decaying facilitation F2. Alternatively, they can be described as manifestations of short- and long-term memory, respectively [42].

The overall mechanism of photocurrent generation and spikes amplification is summarized in Figure 4. The photoexcitation leads to the electrons transition from the valence to the conduction band (1) and the electron-hole recombination occurs spontaneously afterwards (1’). Electrons in the conduction band can be subsequently transferred through the interface to the conducting substrate (2) and holes can migrate to the surface and react with redox mediators in the electrolyte (2’). At the same time a fraction of electrons from the conduction band becomes trapped within interband states in a very fast process (3). This process efficiently competes with the interfacial electron transfer (2), but once the traps are filled this pathway becomes inactive. The trapped electrons undergo relaxation with the time constants τ_1_ (3’) and τ_2_ (3’’).

This simple mechanism provides a platform for the implementation of neuronal dynamics in an artificial, fully inorganic system. Due to its simplicity it can be applied for signal and pattern processing and could be integrated into larger neuromimetic systems. Furthermore, along with bioinspired neuromorphic computing, other information processing paradigms may be implemented within the same system: Boolean logic [43], ternary logic [44] and fuzzy logic [45]. The latter one is especially tempting, as it may contribute to the development of novel neuro-fuzzy information processing devices [46,47,48].

### 2.3. Recognition of Digits

A dataset containing 1000 handwritten digits (100 samples of each 0, …, 9 digit) was randomly selected from the MNIST database (Figure 5a). All the images were transformed into binary strings and used for the modulation of a light source. In order to eliminate possible errors resulting from the photoelectrode equilibration or photodegradation, first 20 and last 20 recorded photocurrent profiles were discarded and the remaining 60 patterns were subjected to further processing. First of all, a set of simple classification rules have been developed. These are based on pixel counting, therefore cannot provide a significant separation of the input data. In the first step, each sign (in a form of 28 × 28-pixel image, Figure 5a) was divided into four quadrants labelled κ_1,_ …, κ_4_ (Figure 5b) and the sum of black pixels confined within each quadrant was calculated. In other words, an individual character was associated with a vector $\left[\Sigma \kappa_{1},\Sigma \kappa_{2},\Sigma \kappa_{3},\Sigma \kappa_{4}\right]$ or a point in 4-dimensional space. Subsequently, four 3-dimensional projections were formulated in the following manner: $\left[\Sigma \kappa_{1},\Sigma \kappa_{2},\Sigma \kappa_{3}\right]$, $\left[\Sigma \kappa_{1},\Sigma \kappa_{2},\Sigma \kappa_{4}\right]$, $\left[\Sigma \kappa_{1},\Sigma \kappa_{3},\Sigma \kappa_{4}\right]$ and $\left[\Sigma \kappa_{2},\Sigma \kappa_{3},\Sigma \kappa_{4}\right]$. For each type of input class (0, …, 9) an ellipsoid with the confidence level of 65% was fitted using 3D Confidence Ellipsoid toolbox in OriginPro 2019.

The collection of data points representing all 600 characters under consideration for $\left[\kappa_{1},\kappa_{2},\kappa_{3}\right]$ combination of quadrants is shown in Figure 6a and the fitted ellipsoids in Figure 6b,c. It can be noticed, that the applied analysis procedure provides a rather poor separation, as the fitted ellipsoids excessively overlap in most of the cases, with the exception of “1” and “9” pair. This result is fully consistent with the initial assumption—simple pixels counting cannot serve as an efficient method for handwritten character recognition.

It can be noticed that only two pairs are completely separated, whereas three others are close to complete separation. Most of these cases concern digit “1” which is substantially different (when the symmetry and number of pixels are taken into account) from any other handwritten digit.

In order to quantify the efficiency of digits recognition in various scenarios, a separability index was defined. Let $V\left(m\left(\kappa_{i},\kappa_{j},\kappa_{k}\right)\right)$ be the volume of an ellipsoid fitted to the digit *m* representation in $\kappa_{i},\kappa_{j},\kappa_{k}$ projection. Then the separability index of the digit *m* to *n* will be defined as a ratio between the relative complement of *n*-ellipsoid in *m*-ellipsoid to the volume of *m*-ellipsoid for *m* ≠ *n* (2):(2)$$\xi_{m/n}=\frac{V\left(m\left(\kappa_{i},\kappa_{j},\kappa_{k}\right)\backslash n\left(\kappa_{i},\kappa_{j},\kappa_{k}\right)\right)}{V\left(m\left(\kappa_{i},\kappa_{j},\kappa_{k}\right)\right)}$$

The calculated separability indices for the input data in one of the possible projections are collected in Table 1. It is noteworthy that the matrix containing separability indices for a given combination of quadrants is not symmetrical, i.e., $\xi_{m/n}\neq \xi_{n/m}$, since ellipsoids have different volumes. If *m* = *n* then $\xi_{m/n}=0$.

In most of the cases the separation is insufficient to allow the unequivocal recognition of handwritten shapes. The exception is a pair {1, 9} for which the separability index is equal to one, as the corresponding ellipsoids do not overlap. It is due to the fact, that “1” differs significantly in terms of pixels distribution between the quadrants. Poor separation can be however greatly improved with the use of even the simplest, the single-node hardware neural network.

In the first step all the characters were converted row-by-row into a stream of bits (“0” for a white pixel and “1” for a black one) and used to modulate the light source according to the scheme presented in Figure 7a,b. The recorded photocurrent spikes (Figure 7c) reflected the sequence of light pulses, but their intensity varied according to the previous states the photoelectrode was in (the short-term memory, vide supra, Figure 3). The photocurrent patterns were subsequently normalized: the amplitude of each signal was divided by the highest intensity recorded for the particular character. The application of various threshold values (Figure 7c,d) acted as a filter for the photocurrent spikes depending on their amplitude. The obtained images with the lowest intensity pixels removed at different thresholds are shown in Figure 8.

The application of different thresholds (from Θ = 0.3, with virtually no signal filtration to Θ = 0.9, corresponding to the removal of all but the most intense pixels) leads to the evolution of the character image, which depends directly on the neighbors of each particular pixel in the row. Significantly, the “distance” (formerly in space, translated into time intervals between the light pulses) from the closest preceding black pixel determines the weight of the subsequent photocurrent spike amplification. As the result, a simple type of classification according to the scattering of pixels can be achieved. Like in the case of the input data, the output images are subjected to the evaluation of respective separability indices at various threshold values. An example is shown in Table 2.

Figure 9a shows a collection of all data points obtained for one selected projection ($\kappa_{1},\kappa_{2},\kappa_{3}$) and one threshold value (Θ = 0.7). They seem to be equally scattered as points for unprocessed data (cf. Figure 6a), but the fitting procedure reveals significant differences. Some ellipsoids, that were initially well separated (e.g., the {1, 9} pair) overlap significantly (Figure 9b). Some other remain unchanged (Figure 9c). More interestingly there are numerous pairs (e.g., {2, 5}, Figure 9d) which are significantly separated upon the data processing with the neuromimetic element.

Upon data treatment with the neuromimetic element the separability is significantly improved. Six pairs of digits are completely separated and two others are close to complete separation. Furthermore, these pairs are different than those separated with the use of the pixel counting method (cf. Table 1).

The improvement of digits classification can be visually evaluated by the comparison of color-coded Table 1 and Table 2. In order to perform a global quantitative evaluation of the separation efficiencies and to assess the improvement associated with the use of the single-node hardware neural device, a separability ratio ($\Xi_{m/n}$) was defined as a ratio of the separability index calculated for the processed data to the separability index determined for the input data (for *m* ≠ *n*) (3):(3)$$\Xi_{m/n}=\frac{\xi_{m/n}^{output}}{\xi_{m/n}^{input}}$$
and $\Xi_{m/n}=0$ for *m* = *n*. Detailed analysis of calculated values provides information on the discussed procedure efficiency even in the case of significantly overlapped ellipsoids. Selected separability ratios for $\kappa_{1},\kappa_{2},\kappa_{3}$ projection at the threshold Θ = 0.7 are presented in Table 3.

It can be noticed that the overall improvement of separability is achieved with at least twofold increase of the index value for 10 pairs compared to the twofold decrease in five instances. A similar situation is also observed for other projections at this threshold. This qualitative picture suggests an improvement of handwritten digits recognition upon application of a neuromimetic element in data processing. A quantitative estimation of classification improvement can be obtained through simple numerical analysis of output data. Due to the analysis complexity (four quadrant combinations for eight different threshold values), various separation scenarios (depending on the chosen threshold and projection) are possible. Their overall efficiency can be evaluated using an integral separability index Ω, which acts as a global parameter indicating performance of the system for all of the above-mentioned variables. For each separation scenario it can be defined as follows (4):(4)$$\Omega \left(\kappa_{i},\kappa_{j},\kappa_{k},\Theta \right)=\frac{\sum_{m}\sum_{n}\xi_{m/n}^{output}}{\sum_{m}\sum_{n}\xi_{m/n}^{input}},m\neq n$$

The above-mentioned dependency of the Ω function values is depicted in Figure 10. It can be noted, that in the majority of investigated separation scenarios the integral separability index after the treatment with the single-node hardware neural network is significantly higher than the value calculated for the unprocessed input data. In three cases the selection of a low threshold value (a situation which results in an insufficient filtration of pixels due to inadequate exploitation of memory features) leads to the inferior separation. On the other hand, when the short-term memory of the system is optimally utilized, the recognition of handwritten digits increases. For two quadrant combinations an optimal threshold value exists, which is fully consistent with the expectations—too deep filtration removes too many pixels and all the data points (cf. Figure 6a and Figure 9a) interfuse at the origin of the coordinate system.

## 3. Conclusions

Surprisingly, even a primitive hardware realization of the neural network architecture based on a single-node exhibiting the short-term memory can significantly improve pattern recognition. Classification based solely on the number of black pixels encompassed by each of the four quadrants the character image is divided into (cf. Figure 5b) is insufficient—only a few characters of a specific symmetry (e.g., “1”) could be distinguished using this primitive procedure. The application of the optoelectronic element with PPF functionality enhances tremendously (for such a simple device) the classification efficiency. This change is based on the extraction of a new feature of the studied data—the scattering of pixels within the 28 × 28 matrix. The high dispersion leads to the negligible photocurrent amplification. On the contrary, the digits with large groups of pixels are characterized by a higher number of counts for relatively high thresholds, for which the short-term memory of the system and resulting photocurrent amplification is strongly pronounced.

The system presented in this work is a single node neural device, the operation of which is based on the unsupervised learning paradigm involving the short-term memory of the device. The simple pixel counting method gives precise information on the size of the characters (therefore digit “1” separates well in all of the cases) and indirectly on their symmetry (it can be achieved by an appropriate selection of quadrant combinations). Application of a neuromimetic element allows further information processing, particularly the extraction of information on pixel “agglomeration”, at least at the single row level. This process is analogous to Sammon mapping [49], but does not involve the reduction of data space dimensionality (Figure 11).

The presented optoelectronic single-node neural device is superior compared to software implementations in terms of the error resistance and the energy efficiency. These features of the discussed system make it a potential low-cost pre-processing unit. Furthermore, due to the operation based on time-series and the intrinsic short-term memory, it can be combined with selected aspects of the reservoir computing paradigm—in one of the possible scenarios, where a delayed feedback loop is used, the virtual neurons could significantly affect the system performance. At the same time, the analog character of the system allows the implementation of the fuzzy logic system, yielding a new class of hardware neuro-fuzzy devices.

The research presented in this paper supports the concept of heterodic computation [27,28]. It clearly shown that the performance of a simple numerical algorithm (classification based on pixel counting) can be improved by in-materio computational component, which itself cannot perform any classification tasks.

## 4. Materials and Methods

Commercially available cadmium sulfide (POCH, Las Condes, Chile), potassium nitrate (Avantor, Radnor, PA, USA), potassium iodide (Aldrich, St. Louis, MO, USA) and iodine (Aldrich, St. Louis, MO, USA) were used as received.

Working electrodes were prepared from polyethylene terephthalate (PET, Camarillo, CA, USA) foil coated with indium tin oxide (Aldrich, St. Louis, MO, USA). These substrates were washed carefully with diluted detergent solution, deionized water and isopropanol. They were then dried in air. The cadmium sulfide was ground with deionized water in an agate mortar to a thick paste and deposited onto the freshly cleaned substrates using a screen-printing machine (MikMetal, Masis St, Yerevan, Armenia) equipped with a 80 mesh polymer grid.

The UV/Vis spectra were recorded using Lambda 750 spectrophotometer (Perkin Elmer, Waltham, MA, USA) within the wavelength range of 200–2000 nm. Barium sulfate of spectral purity was used as a reference material. The X–ray diffraction patterns were recorded with Empyrean (Cu λ_Kα1_ = 1.54060 Å) diffractometer (PANalytical, Malvern, UK) at room temperature with 2θ values ranging from 20 to 80 degrees. The scanning electron images were taken on Versa 3d (FEI, Lausanne, Switzerland) scanning electron microscope operating at 20 kV with an Everhart-Thornley detector. The chemical composition of the CdS sample was confirmed using the energy dispersive X-ray spectroscopy. All electrochemical measurements were performed with the use of SP-300 potentiostat (BioLogic, Cary, NC, USA). Luxeon Star/O Royal Blue diode (465 nm, the total radiometric power of 110 mW) was used as the light source. It was powered through the WA-301 wideband amplifier (Aim-TTI, Cambridgeshire, England). Pulse sequences were generated with TG2512A arbitrary function generator (Aim-TTI, Cambridgeshire, England) triggered with Arduino Uno R3 system.

The photoelectrochemical experiments were performed in air-equilibrated electrolytes using a three-electrode configuration. As the photoactive component the screen-printed CdS electrodes, immersed in an aqueous electrolyte containing 0.1 M KNO_3_, 0.001 M KI and 0.0001 M I_2_, were used. A saturated Ag/AgCl electrode was used as a reference electrode and a platinum wire as a counter electrode. The positive voltage (400 mV) was applied to the working electrode and the photocell was irradiated with short light pulses (300 µs).

The experiment automation was realized based on the program written in Arduino C++ language. All the necessary data processing was performed with the use of programs written in Python 3.7.2.

## Figures and Tables

**Figure 1 molecules-24-02738-f001:**
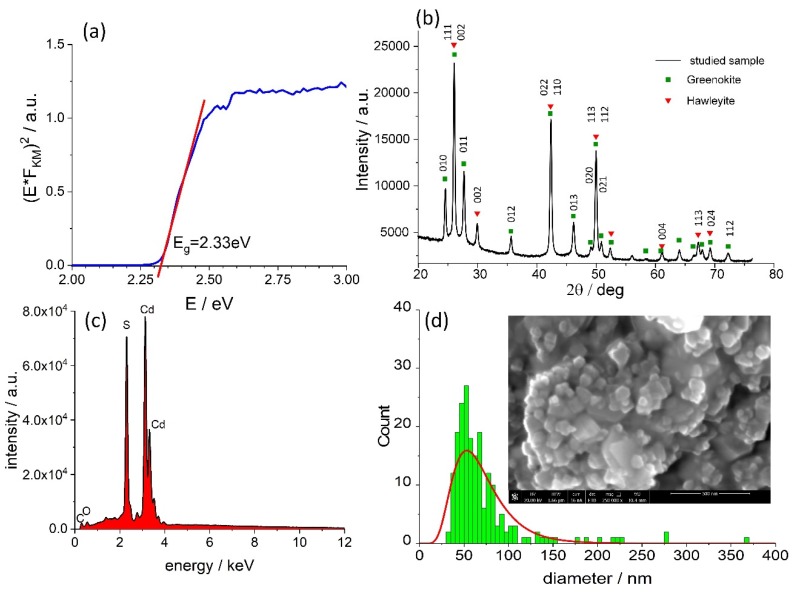
The Tauc plot (**a**), the powder X-ray diffractogram (**b**), the energy-dispersice X-ray spectrum (EDS) (**c**) and the crystallite diameter distribution of the CdS sample (**d**) discussed in this study. Inset shows the SEM image of the studied sample.

**Figure 2 molecules-24-02738-f002:**
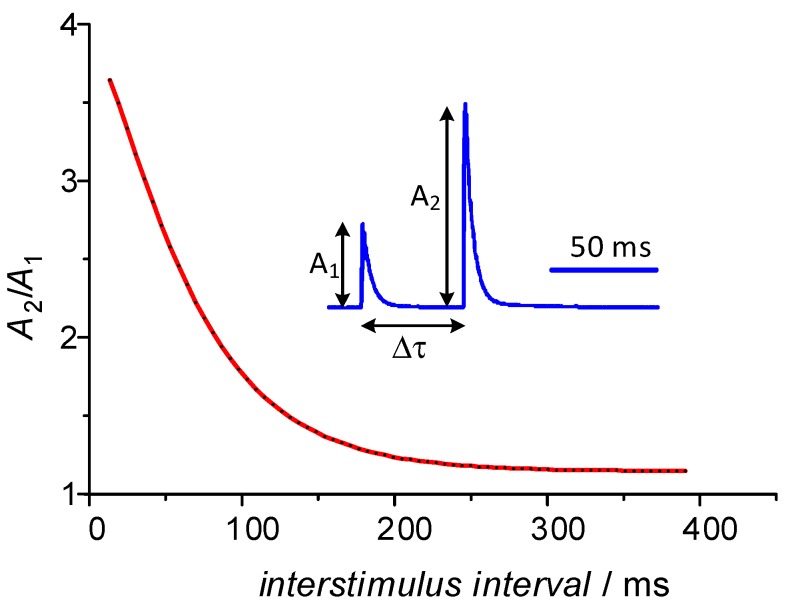
An example of the paired-pulse facilitation observed in nervous system. Adapted from [31].

**Figure 3 molecules-24-02738-f003:**
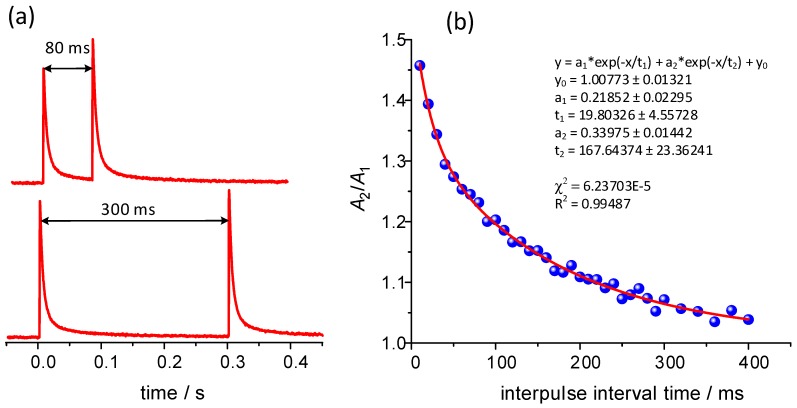
The photocurrent spikes resulting from the pulsed light illumination of CdS-based photoelectrodes (**a**) and the analysis of the photocurrent amplification vs. time interval between subsequent pulses (**b**).

**Figure 4 molecules-24-02738-f004:**
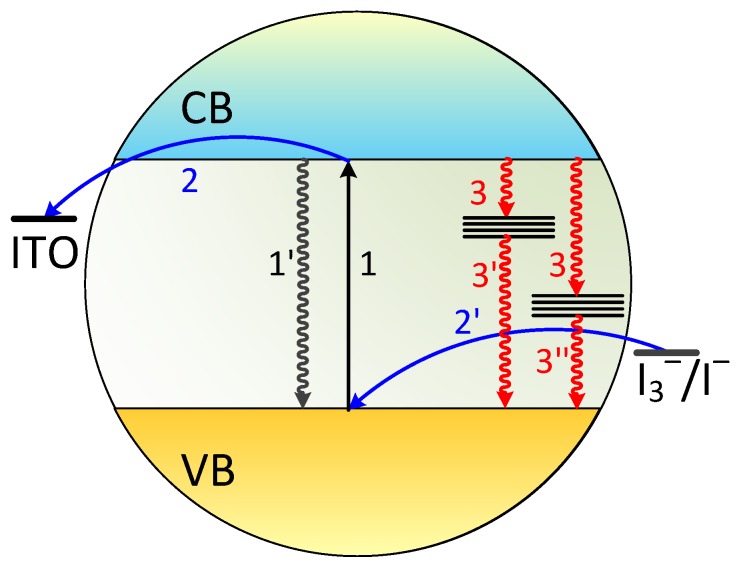
A tentative mechanism of the photocurrent generation and charge carriers trapping in the nanocrystalline CdS sample under consideration.

**Figure 5 molecules-24-02738-f005:**
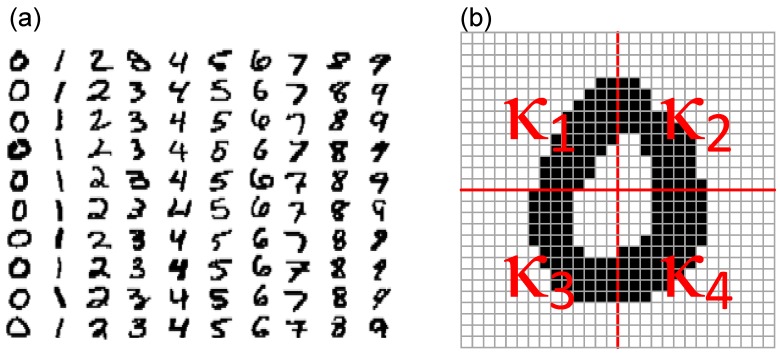
A small sample of the MNIST handwritten digits (**a**) and an image depicting the definition of quadrants for the 28 × 28-pixel image (**b**).

**Figure 6 molecules-24-02738-f006:**
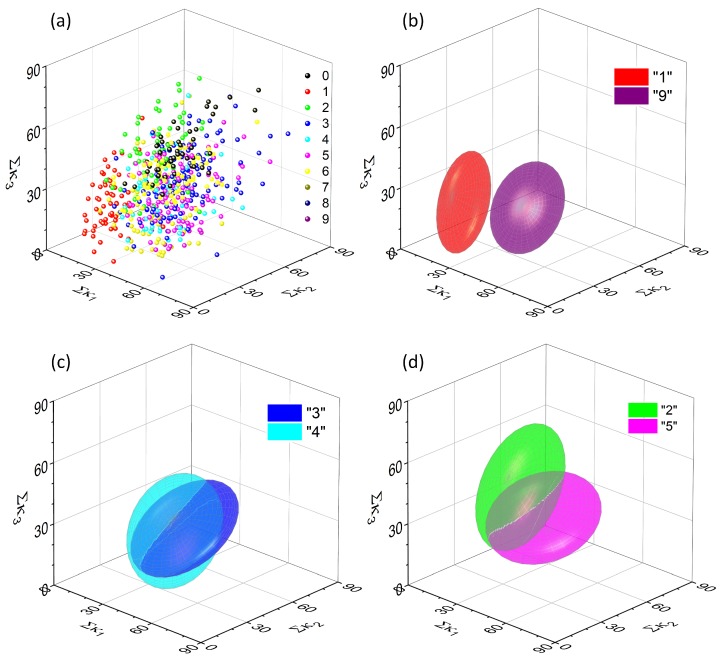
A complete collection of input data points (before feeding them into the single-node neural network) for a set of 600 handwritten digits in the one, arbitrary chosen 3D projection (**a**) and an example of a relatively well-separated pair, which is associated with digits “1” and “9” (**b**). Other ellipsoids overlap significantly, e.g., those for digits “3” and “4” (**c**) or “2” and “5” (**d**).

**Figure 7 molecules-24-02738-f007:**
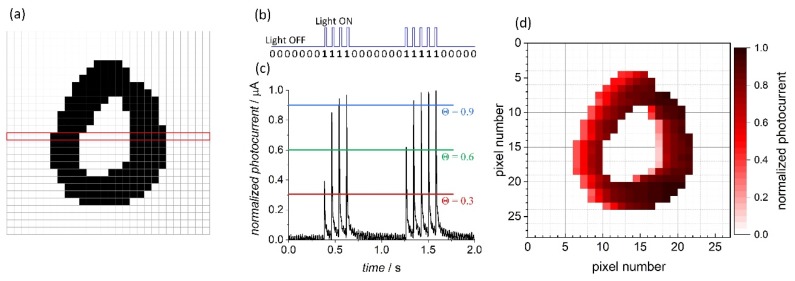
A 28 × 28 pixels image of a handwritten character with a marked row (**a**) translated into a sequence of bits and corresponding light pulses (**b**). A pattern of photocurrent spikes for a given binary input with three thresholds indicated (**c**). An image of the character reconstructed from the normalized photocurrent amplitudes (**d**).

**Figure 8 molecules-24-02738-f008:**
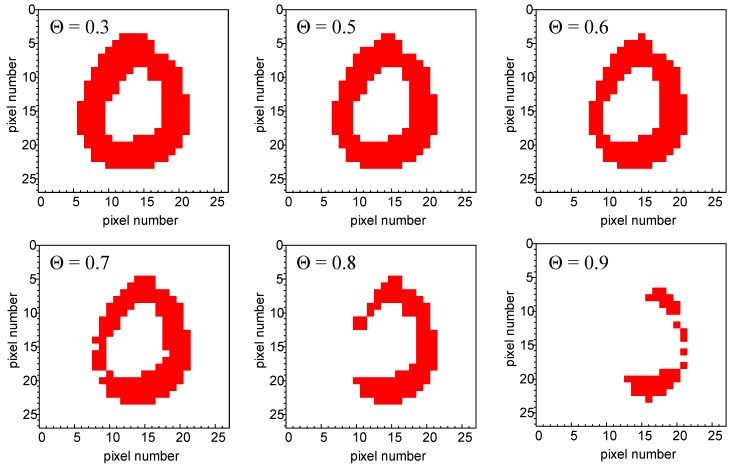
An image of the character from Figure 7 reconstructed from the normalized photocurrent spikes at different threshold values.

**Figure 9 molecules-24-02738-f009:**
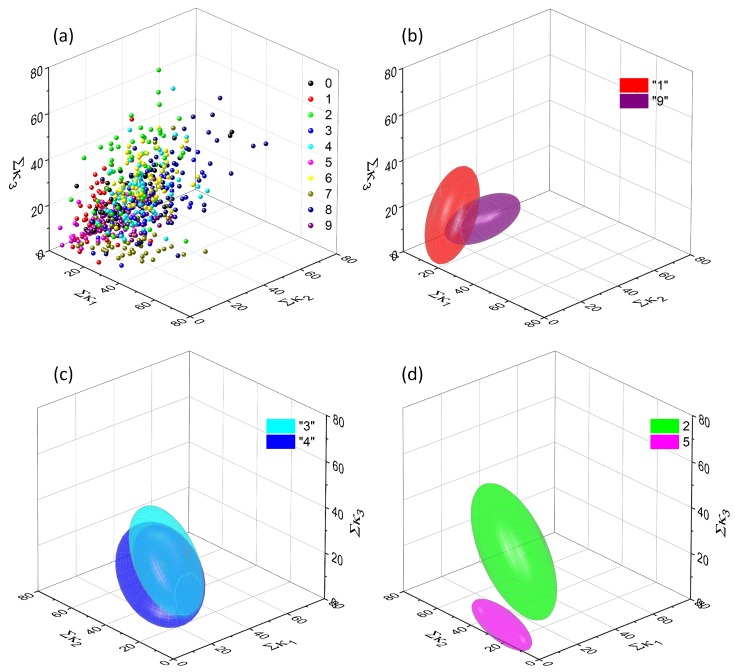
A complete collection of output data points (after feeding them into the single-node neural network) for a set of 600 handwritten digits in the one, arbitrary chosen 3D projection (corresponding to the one shown in Figure 6) (**a**) and an example of significantly overlapped ellipsoids, corresponding to digits “1” and “9” (**b**), “3” and “4” (**c**). The majority of other ellipsoids are separated better, than in the case of untreated data—e.g., those associated with digits “2” and “5” (**d**).

**Figure 10 molecules-24-02738-f010:**
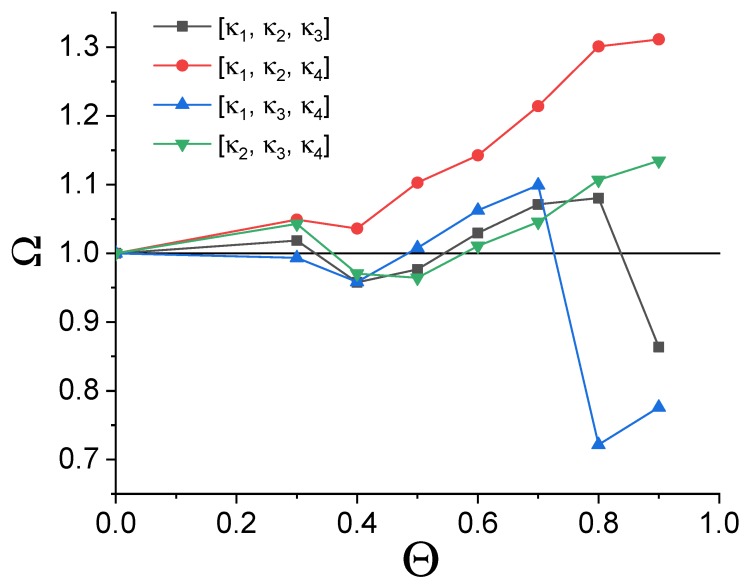
The dependence of the integral separability index Ω vs. the threshold value Θ. The horizontal line indicates the integral separability values determined for the input data. Threshold Θ = 0 corresponds to unprocessed input data.

**Figure 11 molecules-24-02738-f011:**
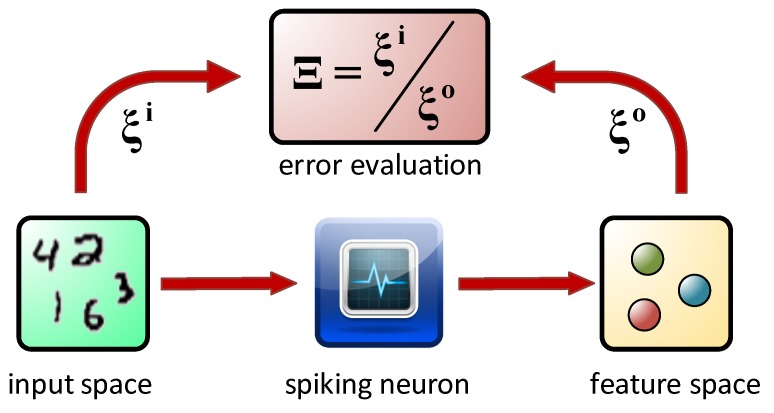
A diagram depicting the data flow and the efficiency evaluation of the unsupervised classification system under consideration.

**Table 1 molecules-24-02738-t001:** The collection of separability indices for the input data in $\kappa_{1},\kappa_{2},\kappa_{3}$ projection. The efficiency of data separation is color coded (vide infra) from red (no separation, *ξ* = 0) to green (perfect separation, *ξ* = 1).

	0	1	2	3	4	5	6	7	8	9
**0**	0.000	1.000	0.753	0.971	0.889	0.812	0.677	0.927	0.838	0.984
**1**	1.000	0.000	0.963	0.977	0.959	0.996	0.982	0.895	0.999	1.000
**2**	0.346	0.900	0.000	0.699	0.556	0.563	0.518	0.617	0.569	0.738
**3**	0.942	0.953	0.770	0.000	0.395	0.360	0.721	0.584	0.523	0.386
**4**	0.781	0.916	0.668	0.406	0.000	0.505	0.685	0.364	0.558	0.080
**5**	0.600	0.991	0.649	0.325	0.469	0.000	0.455	0.689	0.355	0.597
**6**	0.395	0.964	0.659	0.741	0.702	0.520	0.000	0.806	0.692	0.831
**7**	0.841	0.763	0.684	0.550	0.298	0.680	0.773	0.000	0.694	0.179
**8**	0.592	0.998	0.590	0.405	0.438	0.236	0.587	0.648	0.000	0.541
**9**	0.982	1.000	0.890	0.661	0.482	0.789	0.899	0.581	0.797	0.000

**Table 2 molecules-24-02738-t002:** The collection of separability indices for the output data in $\kappa_{1},\kappa_{2},\kappa_{3}$ projection at the threshold Θ = 0.7. The efficiency of data separation is color coded from red (no separation, *ξ* = 0) to green (perfect separation, *ξ* = 1).

	0	1	2	3	4	5	6	7	8	9
**0**	0.000	0.482	0.783	0.548	0.812	0.452	0.923	0.654	0.782	0.366
**1**	0.608	0.000	0.936	0.899	0.969	0.376	1.000	0.808	0.984	0.710
**2**	0.507	0.807	0.000	0.500	0.264	1.000	0.504	0.730	0.653	0.580
**3**	0.457	0.839	0.735	0.000	0.540	0.964	0.878	0.596	0.611	0.364
**4**	0.723	0.940	0.522	0.436	0.000	1.000	0.439	0.801	0.672	0.732
**5**	0.840	0.759	1.000	0.991	1.000	0.000	1.000	0.959	1.000	0.858
**6**	0.896	1.000	0.705	0.863	0.487	1.000	0.000	0.998	0.854	1.000
**7**	0.549	0.669	0.845	0.561	0.824	0.817	0.998	0.000	0.788	0.133
**8**	0.467	0.947	0.626	0.209	0.457	1.000	0.735	0.603	0.000	0.512
**9**	0.670	0.801	0.904	0.725	0.906	0.747	1.000	0.654	0.896	0.000

**Table 3 molecules-24-02738-t003:** The collection of separability ratios ($\Xi_{m/n}$) for the output data in $\kappa_{1},\kappa_{2},\kappa_{3}$ projection at the threshold Θ = 0.7. The efficiency of data separation is color coded from red—significantly decreased separability (Ξ < 0.5), through yellow—slightly decreased separability (0.5 < Ξ < 1), green—moderately improved separability (1 < Ξ < 1.5), blue—significantly improved separability (1.5 < Ξ < 2) to navy blue—outstanding improvement of separability (Ξ > 2).

	0	1	2	3	4	5	6	7	8	9
**0**	0.000	0.482	1.040	0.564	0.913	0.556	1.363	0.705	0.933	0.372
**1**	0.608	0.000	0.971	0.919	1.011	0.377	1.019	0.903	0.984	0.710
**2**	1.468	0.897	0.000	0.715	0.475	1.777	0.974	1.183	1.147	0.787
**3**	0.485	0.881	0.954	0.000	1.369	2.682	1.219	1.019	1.167	0.945
**4**	0.926	1.026	0.781	1.074	0.000	1.979	0.642	2.203	1.205	9.127
**5**	1.398	0.766	1.542	3.055	2.134	0.000	2.198	1.393	2.818	1.438
**6**	2.267	1.037	1.070	1.166	0.694	1.923	0.000	1.239	1.233	1.204
**7**	0.652	0.877	1.235	1.021	2.770	1.202	1.291	0.000	1.135	0.742
**8**	0.788	0.949	1.061	0.515	1.044	4.229	1.253	0.931	0.000	0.947
**9**	0.682	0.801	1.016	1.097	1.877	0.947	1.112	1.125	1.125	0.000

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
