# Peer review of "Hardware Realization of the Pattern Recognition with an Artificial Neuromorphic Device Exhibiting a Short-Term Memory"

_molecules, 2019, doi:10.3390/molecules24152738_

Round 1
Reviewer 1 Report
This report is an interesting study that employed a fully inorganic system exhibiting memory as building block in hardware realizations of the neuromorphic computing.
From my point of view, the description of the material characterization, as well as the description of the chemical system, must be improved:
1) How does the authors explain that the Eg obtained was only for the tetragonal polymorph (2.33 eV) and the one for the hexagonal was not detected? A reference about the literature values will be appreciated for the readers.
2) About the x-ray diffraction, here the authors infer that the material presents both phases. Please provide the corresponding literature reference to validate your results. In addition, the 1:1 relationship that the authors claim is not clear since they did not explain how they determined or compare it. Please provide the Miller indices.
3) Finally, in Line 106, they wrote that the polycrystallinity was revealed by SEM. However, this cannot be inferred from SEM but from X-ray diffraction. In addition, the SEM image showed that the sample is heavy agglomerate, so it makes extremely difficult to detect the mentioned 20 nm particle. Most of the particles seem to be around 80-100 nm. A zoom or another SEM image to show the smallest particles will help to see them. Otherwise, you can also provide a statistic of the particle diameter by Image J software.
4) Please in Fig 3a, would help if you state that one experiment was after 80 ms and the other after 300ns. You can also add in the legend.
5) Figure 3b, can you provide information about the robustness/reproducibility of the experiments. Are they independent experiments or subsequent experiments with the same specimen?
6)In line 141, you indicate that there is a Fig. 3c which is not the draft.
Author Response
1. The difference between the bag gap energies of two polymorphs of CdS is too low to results in two different absorption onsets. Therefore only the lower band gap can be directly observed, as it was previously reported by Zelaya-Angel et al. Literature references and more detailed explanation is added to the text.
2. More detailed description of quantitative analysis of XRD patterns is added to the text. Quantitative analysis was perfomed via deconvolution of diffractogramt using specialized software, which is referenced in the manuscript.
3 .More detailed description, including statistical analysis of SEM images is added to the text as requested.
4. Fig. 3a has a cleas scale indicating time, however for the sale of clarity time intervand were indicated with horizontal arrows and labelled as requested.
5. The data presented in Fig. 3b have been recorded in a series of experiments o single photoelectrode. The material is stable enough to sustain experimental conditions. Obtained values are in good agreement with kinetic photocurrent profiles obtained for similar materials.
6. Typographic errer has been corrected, it should be Fig. 3b.
All changes have been highlighted in yellow.
Reviewer 2 Report
I read with great interest the present submission regarding utilizing spiking induced response of CdS based photoelectrodes for improving the pattern recognition of handwritten numerals database.
The submission is very well written with clear and concise methodology and good overall presentation. Unfortunately the submission lacks the crucial support for the claims made regarding the central result "Hardware realization of the pattern recognition with an artificial neuromorphic device" .
For example: after utilizing the photocurrent spikes as well as best possible threshold selection, the separability indices (shown in Table 2) are not improved over the simple pixel counting method (Table 1). This is further evidenced in color coded table 3 where the red color denoting decreased separability via photocurrent-spike method, is dominant with yellow (slight improvement). The negligibly small fraction of green and blue indicates the poor performance of the method over the simple pixel counting method.
Therefore, with the evidences present, unfortunately I am not convinced about achievement of improved pattern recognition.
Minor yet important point: Page 12 line 319 "The system presented in this work is SOMEHOW similar to the unsupervised neural networks and...." : This is a super weak and negative statement casting doubt over the probable similarity of the system presented with neural networks. Please rephrase the statement and also discuss in few lines how and what is similar and/or dissimilar with neural networks.
Overall, I believe there is merit in the proposed approach, but the required working scheme and supporting data is missing.
Author Response
Tables 1-3 have been described in a more clear way in order to highlight the improved performance of the classification protocol. The color coding of Table 3 has been improved as well to avoid confusion. The number of instances in which classification is improved by 50% is significantly higher than the number of instances when the classification performance is decreased. Along with qualitative descritption based on Tables 1-3 the integral separability index has been redefined and the improvement of
classification in the neuromorphic device is clearly indicated in Fig.
10. For the sake of honesty tables indicate separability indices for an average classification scenario - comparison of all possible scenarios in given in Fig. 10.
2. The final conclusion and discussion has been partially rewritten to avoid confusion and misleading descriptions. The device cannot be regarded as a network as it consists of a single node, details of unsupervised learning has been described in more detail.
All changes have been highlighted in yellow.
Round 2
Reviewer 1 Report
The authors added all the required information, thus improving the manuscript significantly. In my opinion it can be accepted as submitted. Great Job!
Author Response
Dear Editor,
We have introduced further changes, including a short discussion on the concatenation of logic and neuromorphic devices, as suggested by the editor.
With best regards,
Konrad Szaciłowski
Reviewer 2 Report
Thanks to Przyczyna et al. for the revision of submitted manuscript. Unfortunately, the authors have simply rephrased the same statements regarding the success of the method employed, with no added justification.
The earlier comment about improvement of the pattern recognition shown in Table 3 was not about the color scheme, but actual numbers themselves. The authors have simply shifted the goalpost by changing the color scheme. For example earlier green – "moderately improved separability (2 < x < 4)" has become navy blue – "outstanding improvement of separability (x > 2)".
To be more precise the new statement in line 302 sums it up: "In can be noticed that decrease of separability index (40 instances in Table 3 – red and yellow) is significantly less relevant as improved separability cases (50 instances)".
A method which improves recognition for 55% and actually degrades for other 45% is just NO improvement in reality. No method id perfect, but degrading recognition nearly as much improving is not a method "superior compared to software implementations", as claimed here.
I am sorry but, the authors should try either improved device performance and/or photocurrent spike timing to justify improvement over simple pixels counting method.
Author Response
Dear Editor,
I
We believe all the remarks made by Reviewer 2 have been already addressed in the text. Table 3 presents details on the classification results for only one arbitrary chosen (out of 28) scenario and it serves merely to depict the idea behind the analysis. Indeed, it may look at the first glance that the device performance is rather poor, nevertheless, the quantitative analysis (equation 4, Fig. 10 and corresponding explanation in the text) indicates a global improvement in the classification efficiency. We can agree, that there are no perfect systems and the performance of our device is not better than other, more complex and sophisticated solutions, but it is still tremendous taking into account its structural and functional simplicity.
Furthermore, we do not claim that our device is "superior compared to software implementations" as it was pointed by Reviewer 2, but "is superior compared to software implementations in terms of the error resistance and the energy efficiency". Our device is based on a random assembly of nanoparticles deposited onto conducting substrate with the use of the raster printing technique. We are not aware of any computing devices (at least not within classic electronics framework) which work based on a random combination of hardware elements or software instructions. This feature of the presented system contributes to its robustness and almost effortless reproducibility. The other aspect comes from the unsupervised learning approach - the device does not need training, it performs classification on the basis of pixel dispersion within characters and maps the input data space into the feature space (which contains information on the pixel dispersion). This supplements the primitive algorithm (pixel counting) and leads to a significant improvement of the classification procedure (up to 30% when the integral separability index is taken into consideration). Moreover, the discussed system is energy efficient in the sense that molecular/nanoscale devices usually operate without the additional source of energy - the information carriers (light, molecules) provide a sufficient amount of energy to perform computations.
In conclusion, we strongly believe the concepts discussed in the manuscript were misjudged by the Reviewer 2. At the same time, we understand that the text is subject to interpretation, thus we modified the parts regarding the classification improvement in order to clarify the main idea behind the work.
With best regards,
Konrad Szacilowski